# Single-Cell RNA Sequencing Reveals HIF1A as a Severity-Sensitive Immunological Scar in Circulating Monocytes of Convalescent Comorbidity-Free COVID-19 Patients

**DOI:** 10.3390/cells13040300

**Published:** 2024-02-06

**Authors:** Lilly May, Chang-Feng Chu, Christina E. Zielinski

**Affiliations:** 1Leibniz Institute for Natural Products Research and Infection Biology, Department of Infection Immunology, 07745 Jena, Germany; lilly.may@leibniz-hki.de (L.M.); chang-feng.chu@leibniz-hki.de (C.-F.C.); 2Center for Translational Cancer Research (TranslaTUM) & Institute of Virology, Technical University of Munich, 81675 Munich, Germany; 3Department of Microbiology, Friedrich Schiller University, 07743 Jena, Germany

**Keywords:** COVID-19, SARS-CoV-2, immunomonitoring, single-cell RNA sequencing, hypoxia-inducible factor 1-alpha, HIF1A, immunological scar, convalescence

## Abstract

COVID-19, caused by severe acute respiratory syndrome coronavirus-2 (SARS-CoV-2), is characterized by a wide range of clinical symptoms and a poorly predictable disease course. Although in-depth transcriptomic investigations of peripheral blood samples from COVID-19 patients have been performed, the detailed molecular mechanisms underlying an asymptomatic, mild or severe disease course, particularly in patients without relevant comorbidities, remain poorly understood. While previous studies have mainly focused on the cellular and molecular dissection of ongoing COVID-19, we set out to characterize transcriptomic immune cell dysregulation at the single-cell level at different time points in patients without comorbidities after disease resolution to identify signatures of different disease severities in convalescence. With single-cell RNA sequencing, we reveal a role for hypoxia-inducible factor 1-alpha (HIF1A) as a severity-sensitive long-term immunological scar in circulating monocytes of convalescent COVID-19 patients. Additionally, we show that circulating complexes formed by monocytes with either T cells or NK cells represent a characteristic cellular marker in convalescent COVID-19 patients irrespective of their preceding symptom severity. Together, these results provide cellular and molecular correlates of recovery from COVID-19 and could help in immune monitoring and in the design of new treatment strategies.

## 1. Introduction

Coronavirus disease 2019 (COVID-19), caused by severe acute respiratory syndrome coronavirus-2 (SARS-CoV-2), was declared a global pandemic by the World Health Organization (WHO) on 11 March 2020 [1]. Over 772 million cases and over 6 million deaths have been confirmed globally as of November 2023 [2]. COVID-19 is characterized by symptoms such as fever, cough, fatigue and dyspnea [3]. While most patients show a mild or moderate disease course, some patients develop acute respiratory distress syndrome, which in some patients leads to lethal multiple organ failure [4]. Comorbidities such as hypertension, diabetes, chronic obstructive pulmonary disease (COPD), cardiovascular disease and cerebrovascular disease as well as age have been reported to be risk factors for a severe course of COVID-19 [5,6]. However, even young patients without comorbidities are at risk of developing severe symptoms [6,7,8].

The implementation of suitable infection control measures is challenging, especially since some COVID-19 patients are asymptomatic and SARS-CoV-2 infection in these patients is more easily overlooked [9]. Vaccination programs have successfully been introduced, preventing thousands of deaths worldwide [10,11]. Nevertheless, the current vaccines, despite reducing morbidity and mortality, cannot thoroughly prevent infection, especially infection with the new Omicron virus variants [12]. Markers of previous infection, ideally correlated with anamnestic disease severity, would, therefore, be convenient to infer previous COVID-19 characteristics.

Immune dysregulation in COVID-19 patients has previously been examined in several studies. In particular, monocytes have been reported to be an important cell lineage in terms of the immune system’s response to SARS-CoV-2 [13,14]. Transcriptomic studies of COVID-19 patients revealed upregulated cytokine (*TNF*, *CXCL8*, *IL1B*, *IL6*, *IL10*, *IL12*) and chemokine (*CCL2*, *CCL3*, *CCL4*, *CCL20*, *CXCL2*, *CXCL9*, *CXCL10*) expression in monocytes as one major dysregulation in these patients [3,13,15]. Notably, the extent of this upregulation was linked to the severity of a patient’s disease course [3,15]. Others have described a downregulation of major histocompatibility complex (MHC) class II molecules (*HLA-DR*, *HLA-DMA*, *HLA-DPB1*) and some MHC class I molecules (*HLA-E*, *HLA-F*) to be a characteristic feature of COVID-19 [16,17]. This effect has mostly been observed in monocytes. B cells, however, have also been reported to display MHC class II downregulation [17]. Further studies have reported the reduced expression of interferons [18], especially in patients with severe COVID-19 [19]. The importance of interferons in COVID-19 has additionally been highlighted by the finding that patients with defective type I interferon signaling are at higher risk of having a severe or fatal disease course [20].

Upregulation of hypoxia-related genes such as *HIF1A* has been observed in COVID-19 patients [14,21]. It has been shown that COVID-19 patients with hypoxemia and/or dyspnea are at higher risk of dying [22]. Furthermore, persistent hypoxemia and pulmonary hypoxia have been reported in patients with post-acute COVID-19 syndrome, which is characterized by persisting symptoms beyond 4 weeks after symptom onset, with the most common symptoms being fatigue, dyspnea and chest pain [23,24,25]. However, the detailed pathomechanism causing post-acute COVID-19, especially with respect to a role of hypoxia, remains elusive.

Thus, it is necessary to further investigate the molecular mechanisms underlying COVID-19, particularly in patients with no concomitant comorbidities. Since symptoms such as fatigue and anosmia often persist after acute infection, molecular correlates of post-SARS-CoV-2 infection are needed. They might not only serve as retrospective molecular scars of prior COVID-19 disease and of its relative disease severity but also predict future health outcomes.

In this study, we dissected the transcriptomic immune cell dysregulation in six convalescent COVID-19 patients using single-cell RNA sequencing (scRNA-seq). Of note, we recruited our study participants from a unique patient cohort characterized by the absence of relevant comorbidities to exclude the confounding effect of the comorbidity itself.

To track immunologic signatures of divergent disease courses, ranging from asymptomatic to severe disease (requiring hospitalization), symptom severity scores of the individual patients were correlated with their respective single-cell transcriptomic patterns. This revealed HIF1A as a severity-sensitive long-term immunological scar in circulating monocytes of convalescent COVID-19 patients.

## 2. Materials and Methods

### 2.1. Patient Cohort Characterization

Six patients diagnosed with COVID-19 based on a positive SARS-CoV-2 RT-PCR of throat swab samples were assessed after excluding those with pregnancy and preexisting comorbidities (autoimmune diseases, cardiovascular diseases, cancer, lung diseases and therapeutic immunosuppression). Blood samples of each patient were collected 2, 4 and 6 weeks after the positive RT-PCR test result. Additionally, at the 2-week time point, each patient completed a physician-assisted questionnaire assessing 15 of the most common symptoms associated with COVID-19. Each symptom was assigned a score ranging from 0 to 3, with 3 indicating that the respective symptom was most strongly pronounced, resulting in a cumulative COVID-19 severity score ranging from 0 to 45. These symptom scores were more sensitive to differences in the mild to moderate range of disease severity than the WHO COVID-19 Ordinal Scale for Clinical Improvement [26], which covers a broader range of severities for hospitalized patients with a high disease burden. Patients with an Ordinal Scale for Clinical Improvement score ranging from 5 to 8 (WHO), who were previously reported to frequently present with severe comorbidities, were not included into our patient cohort (Appendix A) [3,27].

### 2.2. Cell Isolation and Single-Cell RNA Sequencing (scRNA-seq)

Viable mononuclear cells were sorted from cryopreserved PBMC based on forward/sideward scatter characteristics and by exclusion of Hoechst 33258-positive cells (Appendix A). The total cell count for each sample was 400,000 cells.

Single-cell labeling, cell capture and cDNA synthesis were conducted on a BD Rhapsody Single-Cell Analysis System (BD Biosciences, Franklin Lakes, NJ, USA). The single-cell libraries were prepared according to the manufacturer’s instructions. The BD Rhapsody Targeted mRNA and AbSeq Amplification Kit (BD Biosciences) was used to create the cDNA library. A target primer panel was constructed by combining a commercial predesigned Human T Cell Expression Primer Panel (BD Biosciences, 259 target primers) with our custom-designed primer panel (BD Biosciences), comprising 98 additional genes (Appendix A) for in-depth analysis of immune cell lineage annotations, T cell residency, immune cell metabolism and immune functions.

Target cDNA was amplified in 11 PCR cycles using this combined primer panel. PCR1 products were purified with AMPure beads (Beckman Coulter, Krefeld, Germany). Next, the mRNA target products and sample tag products were size-separated with 0.7- and 1.2-fold bead-product ratios, respectively. The purified and separated PCR1 products were amplified in 10 PCR cycles, yielding PCR2 products, which were then purified with AMPure beads. The PCR2 mRNA target and sample tag products were then size-separated with 0.8- and 1.2-fold bead–product ratios, respectively. The concentration of the resulting PCR products was estimated using a Qubit Fluorometer (Thermo Fisher Scientific, Dreieich, Germany) with a High Sensitivity dsDNA Kit (Thermo Fisher Scientific). Final products for library index PCR were diluted with elution buffer (BD Biosciences) into 2 ng/μL of mRNA targeted PCR2 product and 1 ng/μL of sample tag PCR2 product. PCR2 products were amplified in 6 PCR cycles using index primers for the final library preparation. The quality control of the final library products was performed using the Qubit Fluorometer and the Agilent 2100 bioanalyzer with a High Sensitivity Kit (Agilent, Waldbronn, Germany). Library products were pooled to a final concentration of 4 nM corresponding to an mRNA/sample tag ratio of 90/10%. Twenty percent of PhiX control DNA was spiked into the final pooled library, which was then sequenced on a NovaSeq platform (Illumina, München, Germany).

### 2.3. scRNA-seq Data Analysis

A count matrix was generated from the sequencing files with the BD Rhapsody Analysis pipeline on the Seven Bridges Genomics platform (BD Biosciences). In total, the count matrix consisted of 86,317 cells and 357 genes. Further preprocessing steps as well as the analysis were performed using the scanpy [28] package (version 1.8.2) for Python (version 3.8). Cells identified as “Undetermined” or “Multiplet” by the BD Rhapsody Analysis pipeline were filtered out, leaving 80,147 cells. Only genes expressed in at least 10 cells were included in the downstream analysis. A total of 342 genes met this criterion. Subsequently, low-quality cells, defined as having a unique molecular identifier (UMI) count below 50 (1575 cells) or greater than 1250 (52 cells), were removed, leaving 78,520 cells.

Counts per cell were normalized (scanpy.pp.normalize_total()) and the results were subsequently logarithmized (scanpy.pp.log1p()). Afterwards, batch correction was performed between patient samples with assigned clinical scores 0 and 17 and the other four patients, using the Combat [29,30] algorithm (scanpy.pp.combat() method) (Appendix A). Next, dimensionality reduction using principal component analysis (PCA) (scanpy.tl.pca() with n_comps = 30) was performed. Following the approach that principal components (PCs) should account for at least 75% of the total variance [31], the first ten PCs were used, explaining 75% of the total variance (Appendix A), to calculate a neighbor graph (scanpy.pp.neighbors() with n_pcs = 10 and default parameters otherwise). For visualization purposes, the neighbor graph was then embedded in two dimensions using the uniform manifold approximation and projection (UMAP) algorithm [32] (scanpy.tl.umap()). The calculated neighbor graph was clustered using the Leiden algorithm [33] (scanpy.tl.leiden()) with a resolution of 1.5 (Appendix A). Different values for the resolution parameter, controlling the granularity of the clustering, were qualitatively assessed based on the identifiability of the cell lineages. Cell types were assigned to each cluster based on the expression of cell-type-specific marker genes according to previously published reports [34]. Regulatory T cells (Treg cells) were annotated as cells expressing Treg marker genes (*CTLA4*, *IL2RA*, *ICOS*, *FOXP3*) (Appendix A). They clustered together with conventional CD4^+^ T cells. Treg cells within the CD4^+^ T cell cluster were identified by setting a threshold (6.75) on the summed expression of all four marker genes (Appendix A).

### 2.4. Statistics

The Wilcoxon rank sum test (scipy.stats.ranksums()) was used to compare gene expression in different cell subpopulations and to identify significance. For comparison of gene expression distribution over time, a Wilcoxon rank sum test with the alternative hypothesis that the expression at the respective earlier time point is greater than the expression at the later time point was performed. The correlation between gene expression and the clinical score of a patient was assessed based on Pearson’s correlation coefficient (PCC). We further computed Spearman’s rank correlation coefficient ρ to identify potential co-expression between different genes because this measure has been shown to effectively identify associations between genes in comparison to other indicators, such as the PCC [35]. The change in gene expression over time was further investigated using the log2-fold change (log2FC), which was calculated by dividing the mean log-normalized expression of the respective earlier time point by the mean log-normalized expression of the later time point and subsequently taking the log2 of the quotient. *p* values of 0.05 or less were considered significant.

## 3. Results

### 3.1. The Immune Cell Composition of Convalescent COVID-19 Patients Demonstrates Interindividual Stability over Time

To identify immune abrogations indicative of prior COVID-19 disease or long-term remnants after symptom resolution in convalescent patients, we investigated PBMCs of convalescent COVID-19 patients with a range of disease severities using scRNA-seq over time. We investigated six convalescent COVID-19 patients (five males, one female) who were specifically selected on the basis of the absence of any comorbidity, which represents one of the strongest confounding factors in immunomonitoring and in COVID-19 disease severity scoring [5,36]. We also focused on patients with a mild to moderate disease course to identify signatures that would be applicable to the majority of the world population during the pandemic and that would be sensitive enough to discern differences within this rather homogenous patient population. The age of the patients ranged from 29 to 65 (mean age 49). Blood samples of each patient were collected 2, 4 and 6 weeks after a PCR-test-validated SARS-CoV-2 infection (Figure 1a).

Patients were assigned a symptom severity score ranging from 0 to 45. The score accounts for 15 individual COVID-19-associated symptoms. The severity scores of the six study patients ranged from 0 (asymptomatic) to 26 (moderate-severe). This matched a COVID-19 severity ranging from 1 (ambulatory, no limitation of activities) to 4 (hospitalized, mild disease) on the WHO COVID-19 Ordinal Scale for Clinical Improvement (total range: 0–8) [26] (Appendix A).

We next assessed whether the immune cell composition upon clinical recovery was altered over time after COVID-19 disease in correlation to different disease severities. To this end, we first annotated the various immune cell lineages based on the differential expression of their respective marker genes as described previously [34]: CD4^+^ T cells (*CD3D*, *CD3E*, *TRAC*, *TRBC2*, *CD4*, *IL7R*), CD8^+^ T cells (*CD3D*, *CD3E*, *TRAC*, *TRBC2*, *CD8A*, *CD8B*), γδ T cells (*TRDC*), NK cells (*GNLY*, *GZMB*, *NKG7*), NK T cells (*KLRB1*), B cells (*MS4A1*), dendritic cells (*IRF4*) and monocytes (*LGALS3*) (Figure 1b,c). Treg cells were identified with the marker genes *CTLA4*, *IL2RA*, *ICOS* and *FOXP3* (Appendix A). They represented 8.8% of all CD4^+^ T cells based on this annotation, but did not cluster separately from conventional CD4^+^ T cells (Appendix A).

The immune cell composition at different time points post COVID-19 disease was compared to healthy reference values. We found that the various innate and adaptive immune cell frequencies in convalescent patients were within their respective physiological range or deviated only slightly. Furthermore, we found longitudinal stability in the relative frequency of all cell types (Figure 1d and Appendix A). We complemented this analysis with high-dimensional flow cytometric analysis of PBMCs. As expected, there was some variation in cell type frequencies depending on whether transcriptomic or protein-based cell type assignment was performed. The cell type assignment based on the transcriptome, for instance, resulted in fewer frequencies of αβ T cells but higher frequencies of monocytes (Figure 1e). Treg cells were detected by flow cytometry according to the multi-parameter gating strategy as CD4^+^CD25^high^CD127^low^ T cells (Appendix A). Transcriptomic identification could not discern Treg cells as a separate cluster differing from conventional CD4^+^ T cells, but revealed their identity by marker gene expression (Appendix A). Importantly, we demonstrated that the overall immune cell composition, whether assessed on a single-cell transcriptomic or flow cytometric level, did not substantially differ with respect to prior COVID-19 disease severity. The stability of the immune composition could suggest that the quantitative analysis of immune cell lineages is not sensitive enough to reveal immunological scars that correlate with previous COVID-19 disease severity. This stresses the need for more in-depth functional single-cell analyses to reveal a potential immune dysregulation in convalescent COVID-19 patients.

### 3.2. Transcriptomic Dysregulations Typical for COVID-19 Are Absent in Early Convalescent Patients

Previously, an upregulation of cytokine and chemokine expression in monocytes of COVID-19 patients was reported [3,15,17]. Additionally, MHC class I and II molecules have been shown to be downregulated in antigen-presenting cells such as monocytes and B cells [16,17]. Whether these disease-associated immune cell alterations persist in recent convalescent patients and would, therefore, provide an immunological scar despite complete disease resolution remains unknown. Therefore, the expression of MHC class I and II molecules, cytokines and chemokines was investigated longitudinally (2, 4 and 6 weeks after validation of infection) in six convalescent COVID-19 patients with different disease severities.

We examined the expression of a range of cytokine and chemokine genes known to be associated with the myeloid cell lineage in monocytes [37]. We identified altered expression of *IL1B*, *CXCL8* (*IL8*) and *CCL3*. When examining the samples of all six patients together, these three genes showed upregulated expression in week 2 compared to the other time points. All other investigated pro- and anti-inflammatory cytokines and chemokines exhibited low expression levels, with only a small number of cells expressing the respective genes (Figure 2a). This excludes that a cytokine storm caused by monocytes in convalescent COVID-19 patients extends to the early convalescent phase of COVID-19.

The genes *IL1B*, *CXCL8* and *CCL3*, which we identified to be significantly upregulated during early convalescence, are known as proinflammatory markers of acute COVID-19 disease [38]. We, therefore, investigated the expression of these three genes in each patient individually. For *IL1B*, only three out of the six patients exhibited a significant upregulation in week 2 compared to both later time points, with no clear association between upregulation and disease severity. Additionally, for *CXCL8* and *CCL3*, the expression profile was neither consistent across all patients nor was there an association with the clinical score of the respective patient (Figure 2b). We conclude that the observed upregulation of *IL1B*, *CXCL8* and *CCL3* in week 2 compared to the later time points is attributable to interindividual patient differences rather than a general feature of convalescent COVID-19 patients.

Previous studies reported a downregulation of MHC class II and some class I molecules in COVID-19 patients [16,17]. We, therefore, investigated the expression of MHC class I and II molecules in monocytes and B cells. When examining the expression profile of each patient in our cohort individually, we identified longitudinal stability in MHC molecule expression in monocytes as well as in B cells (Figure 2c). This implies that there is no alteration in MHC molecule expression levels in convalescent COVID-19 patients. Differences in MHC expression among differentially affected individuals instead suggested a reflection of their interindividual profiles. Taken together, most transcriptomic dysregulations known to be characteristic of SARS-CoV-2 infection do not persist in convalescent COVID-19 patients.

### 3.3. HIF1A Expression Is Significantly Upregulated in Monocytes of Symptomatic Patients Two Weeks after COVID-19 Disease Verification

After having explored major COVID-19-associated immune perturbations in our patient cohort and after having excluded their impact on the early convalescent immune profile, we set out to identify, in an unbiased manner, specific markers that would be indicative of the severity of the preceding disease course. To this end, we investigated the differentially expressed genes at each time point compared to the respective two other time points. Interestingly, our longitudinal study identified *HIF1A* as one of the top upregulated genes in week 2 compared to weeks 4 and 6 (Figure 3a).

*HIF1A* has previously been shown to induce proinflammatory responses through proinflammatory cytokines, and to serve as a positive regulator of SARS-CoV-2 replication [39,40]. Our observation of upregulated *HIF1A* expression in convalescent patients, therefore, prompted a more in-depth investigation. First, we aimed to reveal the respective cell type(s) contributing to the increased *HIF1A* expression. With that goal, we considered the log-normalized *HIF1A* expression in a UMAP representation of all cells, which depicted monocytes as the cell type with the highest *HIF1A* expression (Figure 3b). Additionally, 90.24% of the monocytes had *HIF1A* expression greater than zero, which was the highest percentage of *HIF1A*-transcript-positive cells among all cell types (Figure 3c). Of note, when exploring the *HIF1A* expression in the Treg cell subset separately from the CD4^+^ T cells, we found that only 51.78% of the Treg cells expressed *HIF1A*, a percentage substantially smaller than that observed for monocytes (Appendix A). The importance of monocytes for *HIF1A* expression was further corroborated by a longitudinal examination of *HIF1A* expression in each of the annotated cell types separately. Only B cells, dendritic cells and monocytes showed a substantial number of cells with a log-normalized *HIF1A* expression well above zero (Figure 3d).

Multiple immune cell lineages also showed altered *HIF1A* expression at week 2 compared to later time points, despite containing quantitatively fewer *HIF1A*-expressing cells compared to monocytes. *HIF1A* expression in week 2 was significantly upregulated compared to that in week 4 in CD4^+^ T cells, CD8^+^ T cells, NK cells, monocytes and B cells. All analyzed cell types except for NK T cells and γδ T cells expressed *HIF1A* at significantly higher levels in week 2 than in week 6. In addition, a significant upregulation was observed in CD8^+^ T cells, NK cells and dendritic cells in week 4 compared to week 6 (Figure 3d).

In sum, these findings suggest that *HIF1A* upregulation outlasts acute SARS-CoV-2 infection despite the absence of clinical symptoms of COVID-19 and absence of previously described transcriptomic dysregulations characteristic of acute COVID-19 such as those related to a cytokine storm or MHC downregulation. Moreover, monocytes were identified as the main contributing cell type for this differential expression of *HIF1A*. Additionally, there was a substantial subpopulation of B cells with elevated *HIF1A* expression in week 2.

To test a possible association between *HIF1A* expression and disease severity, we examined its expression in monocytes and B cells for each patient individually. Monocytes and B cells from the patient with asymptomatic COVID-19 (clinical score of 0) showed no significant upregulation of *HIF1A* in week 2 compared to both later time points. The expression of *HIF1A* in monocytes of the other five patients showed significant upregulation in week 2 compared to week 4. Additionally, a significant upregulation of *HIF1A* in week 2 compared to week 6 was observed in the monocytes of four of those five patients (patients with clinical scores of 11, 17, 19 and 26, respectively). For the B cells, only the patients with clinical scores of 17 and 26 showed significantly upregulated *HIF1A* expression in week 2 compared to later time points (Figure 3e). Taken together, a significant upregulation of *HIF1A* expression was revealed in monocytes and in a few patients also in B cells during convalescence. Notably, this was only the case in patients who had previously exhibited a more severe COVID-19 disease course. This leads to the conclusion that *HIF1A* serves as not only an immunological scar of SARS-CoV-2 infection but also a retrospective indicator of previous COVID-19 disease severity in convalescence.

### 3.4. HIF1A Upregulation during Convalescence Significantly Correlates with Previous COVID-19 Disease Severity Scores

Building upon the significantly upregulated *HIF1A* expression observed exclusively in previously symptomatic COVID-19 patients, we further hypothesized that the degree of *HIF1A* upregulation in week 2 could be associated with the severity of former symptoms. To test this hypothesis, we investigated the log2FC of the mean log-normalized *HIF1A* expression in each cell type. For each patient, we compared the expression of *HIF1A* in week 2 to weeks 4 and 6. We identified a strong positive correlation between the clinical score describing symptom severity and the change in the mean *HIF1A* expression in monocytes, B cells and CD4^+^ T cells. In CD4^+^ T cells, the change in *HIF1A* expression from week 2 to week 4 correlated significantly with the clinical score (Figure 3f). When examining Treg cells separately from CD4^+^ T cells, we did not observe a significant association between clinical score and change in *HIF1A* expression (Appendix A). In B cells, the clinical score demonstrated a significant correlation with the change in *HIF1A* expression at week 2 compared to week 6. Monocytes, identified as the primary cell type contributing to the upregulated *HIF1A* expression at week 2, showed a significant correlation between the clinical score and the altered expression from both week 2 to week 4 and week 2 to week 6 (Figure 3f).

This analysis of the change in *HIF1A* expression over time confirms our hypothesis that the degree of the upregulation of *HIF1A* in monocytes in early convalescence (2 weeks after disease resolution) is linked to the severity of previous COVID-19 disease symptoms. Therefore, *HIF1A* expression by monocytes could serve as a biomarker for the severity of COVID-19 even in convalescent patients without residual disease symptoms 2 weeks after infection.

Monocytes represent a cell population that is heterogeneous in terms of phenotype and function [41]. We excluded the possibility of *HIF1A* enrichment within a specific monocyte subset by visualizing all cells with the assigned cell type “Monocyte” using UMAP after calculating a neighbor graph. We found that *HIF1A* expression was homogenously distributed (Appendix A), and clustering was mainly associated with interindividual differences between patients (Appendix A). We concluded that high *HIF1A* expression is a general feature of all monocytes in recently recovered COVID-19 patients and cannot be attributed to a specific dysregulated subpopulation.

Next, we tested the gene regulatory network of *HIF1A*. We did not identify a strong co-expression of *HIF1A* and other tested genes (for all genes, ρ < 0.5 and > −0.4). The genes exhibiting the highest correlation with *HIF1A* expression were *CXCL8* (ρ ≈ 0.48) and *TLR2* (ρ ≈ 0.47) (Figure 3g).

To additionally investigate *HIF1A* expression in healthy controls and to validate our results across a larger patient cohort, we took advantage of a publicly available scRNA-seq COVID-19 dataset that matched our study setup in terms of having multiple sampling time points. The selected dataset [42] comprises COVID-19 patient samples collected longitudinally throughout the disease course and the convalescence stage. In addition, the patient cohort exhibits varying degrees of severity (moderate, severe and critical), although the assessment of symptom severity is not as finely graded as in our study, and confounding factors due to therapeutic measures are possible. Importantly, the public dataset also includes healthy control patients, therefore, not only enabling the verification of our findings so far but also allowing for a comparison of *HIF1A* expression between COVID-19 and healthy subjects [42].

To align with our study setup, wherein the patient cohort is characterized by absence of comorbidities and, consequently, absence of a critical disease course, we excluded samples with a ‘critical’ severity from our analysis of the publicly available dataset. Subsequently, we explored the longitudinal *HIF1A* expression in monocytes of moderate and severe patients, respectively. At each sample time point, we examined whether the *HIF1A* expression was upregulated compared to that in healthy control patients. No significant upregulation was observed in the convalescent stage (14 days or more since disease onset) for moderate patients compared to healthy controls, aligning with our finding that there is no significant *HIF1A* upregulation in patients with a low symptom severity score. In contrast, for severe patients, we observe a significant upregulation compared to healthy controls in the convalescent stage for all time points except one (Appendix A). Overall, this extended analysis corroborates our findings. It supports the conclusion that *HIF1A* acts as a severity-sensitive immunological scar in convalescent COVID-19 patients.

In summary, our data suggest increased *HIF1A* expression to persist as an immunological scar in monocytes and in B cells, with upregulated expression approximately 2 weeks after SARS-CoV-2 infection compared to later time points despite the absence of symptoms or other major transcriptomic dysregulation at that time point. Notably, *HIF1A* upregulation in monocytes persisted beyond an individual monocyte’s half-life, suggesting SARS-CoV-2-induced abrogation of the monocyte precursor niche. Importantly, we also found a correlation of the degree of *HIF1A* upregulation and the severity of previous COVID-19 symptoms.

### 3.5. Circulating Monocyte–T Cell and Monocyte–NK Cell Complexes Indicate a Persistent Dysregulation in Convalescent COVID-19 Patients

In addition to the enrichment of *HIF1A* in the monocyte subpopulation, a second cluster depicting high *HIF1A* expression was observed within the UMAP (Figure 4a). This subcluster consisted of 1137 cells (1.45% of all cells) and contained cells from all patients as well as all time points. No correlation between COVID-19 disease severity and the number of cells of the respective patient in the cluster could be identified. A total of 93.14% of the cells in that population expressed *HIF1A*, which even exceeded the percentage of *HIF1A*-expressing cells in the monocyte population.

No classic cell type could be assigned to the cell population using cell type annotation with marker genes. Strikingly, the cells in question expressed marker genes typical of T cells (*CD3D*, *CD3E*, *TRAC*, *TRBC2*), NK cells (*GNLY*, *NKG7*) and monocytes (*LGALS3*) (Figure 4b) [43]. For a more in-depth characterization of the cell population, the expression of selected marker genes was examined in the UMAP representation of these cells after calculating a neighbor graph. The monocyte marker gene *LGALS3* was homogenously expressed over the whole UMAP. A subcluster with little to no T cell marker gene (*CD3D*) and high NK cell marker gene (*GNLY*, *GZMB*, *NKG7*) expression was identified. Additionally, we noted heterogeneity within the subset of *CD3D*-expressing cells. Distinct subsets expressing CD4^+^ T cell (*IL7R*), CD8^+^ T cell (*CD8A*) and γδ T cell (*TRDC*) marker genes were identified (Figure 4c). We, therefore, hypothesized that the identified cluster likely consisted of monocytes forming complexes with NK cells and T cells. Notably, a *CD14*^+^ *CD3*^+^ double-positive subset was described in a study with convalescent COVID-19 patients before [43], which aligns with our observation. Additionally, previous research suggests that the presence of T cell–monocyte complexes indicates immune perturbation [44].

Since cellular complexes and doublets should display an increased number of mRNA molecules, we tested the number of unique molecular identifiers (UMIs) per cell. Consistent with this assumption, the UMI count in the subcluster of interest was significantly increased compared to all other cell types (Figure 4d; *p* value of Wilcoxon rank sum test: 4.36 × 10^−123^, alternative hypothesis: UMI counts in subcluster of interest greater than those of monocytes).

We then investigated whether this subcluster displayed disease-severity-sensitive *HIF1A* expression, as observed in monocytes by examining *HIF1A* expression in each patient over time. We identified a significant *HIF1A* upregulation in week 2 compared to both later time points in two patients. Two additional patients showed a significant upregulation in week 2 over, respectively, one later time point. The asymptomatic patient, on the other hand, did not show any significant *HIF1A* upregulation in week 2 compared to later time points (Figure 4e). We then correlated the change in *HIF1A* expression in the T cell–monocyte cluster over time (log2FC) with COVID-19 severity scores. We identified a moderate to strong correlation (PCC ≈ 0.56 and PCC ≈ 0.8) [45]; however, the correlation was not significant (*p* ≈ 0.25 and *p* ≈ 0.06) (Figure 4f).

Taken together, the significantly higher UMI count as well as the ambiguous marker gene expression leads to the conclusion that monocytes forming cell complexes with either T cells or NK cells are characteristic of COVID-19 patients in early convalescence. These cell clusters displayed a severity-dependent *HIF1A* upregulation in week 2.

## 4. Discussion

In this study, we investigated molecular correlates of COVID-19 disease and its severity in a cohort of convalescent patients. This revealed that specific markers persist in the blood after disease resolution, in particular, *HIF1A* expression within monocytes, and thus represent immunological scars that are indicative of the previous disease course despite the absence of concomitant symptoms.

ScRNA-seq enables the dissection of cellular heterogeneity with unprecedented resolution [46,47]. This allows for the identification of cellular and molecular correlates of disease pathogenesis and disease resolution. In this study, we aimed to identify whether convalescent COVID-19 patients carry cellular signs of transcriptomic dysregulation indicative of previous disease severity. In contrast to most previous studies investigating COVID-19 patients using scRNA-seq [48,49], we took samples from convalescent patients and followed them over several time points in each individual patient, allowing for a longitudinal investigation. Additionally, we excluded patients with pre-existing medical comorbidities to exclude their impact on the immune status, which distinguishes our research from previous publications [17,50].

We revealed that *HIF1A*, the gene encoding a key regulator of a cell’s response to hypoxia [51], can serve as an immunological scar in monocytes of patients who previously recovered from SARS-CoV-2 infection. Furthermore, the disease severity significantly correlated with the degree of *HIF1A* upregulation early after full recovery from COVID-19. While a patient with asymptomatic COVID-19 did not show a significant upregulation of *HIF1A* at week 2, the magnitude of the upregulation increased in patients with a more severe disease course. This finding is especially striking considering the innate nature as well as the short half-life of monocytes. Monocytes circulate in the blood for only one to three days before entering peripheral tissues or dying. Only a small fraction of monocytes remain in the blood for a longer time period (up to 7.4 days) [52]. The finding of upregulated *HIF1A* expression during convalescence, therefore, suggests some alterations in the replenishment of the monocyte pool from precursors in the bone marrow. Since alterations in hematopoiesis have previously been observed in response to SARS-CoV-2 infections, particularly in individuals with severe COVID-19 [53,54], it can be speculated that this leaves a longer-lasting impact on the myeloid lineage. *HIF1A* expression by monocytes after disease resolution, therefore, provides a retrospective molecular marker of COVID-19 disease and its disease severity.

SARS-CoV-2 enters a cell via angiotensin-converting enzyme 2 (*ACE2*) [55], which has previously been shown to be downregulated by *HIF1A* [56]. Therefore, the upregulation of *HIF1A* has been proposed as a potential mechanism for reducing the invasiveness of SARS-CoV-2 [40]. However, little to no *ACE2* expression has been reported in immune cells from human peripheral blood [57,58]. Yet, activated and alveolar tissue macrophages, which originate from circulating blood monocytes, display high *ACE2* expression and are thus likely to be targeted by SARS-CoV-2 [57]. The upregulated *HIF1A* expression in peripheral monocytes found in this study could potentially induce *ACE2* downregulation, thus hindering SARS-CoV-2 entry into macrophages. It can be speculated that this mechanism would represent an effective strategy of the patient’s immune system to prevent further infection of host cells, thus outlasting acute COVID-19 disease into the convalescent phase.

We have identified a cluster of monocytes forming complexes with either T cells or NK cells. In active tuberculosis and dengue fever, circulating T cell–monocyte complexes have been described before to be a sign of immune perturbation [44]. Cell complex formation has previously been shown to be associated with the disease severity of the aforementioned infectious diseases [44]. It is, therefore, warranted to take special care with doublet exclusion, which is a standardized approach in scRNA-seq analysis pipelines, to avoid overlooking this biologically relevant information. Although no significant correlation between COVID-19 disease severity scores and the presence of monocyte–T cell or monocyte–NK cell complexes could be established, the presence of circulating complexes still indicated an immune perturbation in convalescent COVID-19 patients. Importantly, our finding is consistent with another previous report of circulating *CD14*^+^ *CD3*^+^ double-positive cells in convalescent COVID-19 patients [43]. In this study, we observed high *HIF1A* expression within this unconventional subpopulation. This was especially noteworthy considering that *HIF1A* is known to mediate the upregulation of adhesion molecules that promote cell–cell interactions in hypoxic conditions [59,60]. Therefore, it is tempting to speculate that the formation of monocyte–T cell and monocyte–NK cell complexes could be mechanistically associated with the upregulation of *HIF1A* in the monocytes of convalescent COVID-19 patients.

Although this study provided an in-depth analysis through single-cell transcriptomics of clinically well-characterized patients without comorbidities over time at three distinct time points, respectively, this study was still small in size. Larger studies comprising more than the six included patients are, therefore, warranted to generalize our findings for a large patient population. Nonetheless, a reanalysis of data generated by other studies supports our major conclusion that *HIF1A* serves as a robust retrospective marker for COVID-19 disease severity [42]. Additionally, animal models might further serve to dissect the role of *HIF1A* as a retrospective marker of COVID-19 disease severity. Furthermore, our work raises the question of whether other viral infections would also translate into sustained abrogation of innate immune parameters and whether they would then also serve as retrospective readouts of prior disease severity.

Taken together, this study has identified *HIF1A* as an immunological scar of COVID-19, which is sensitive to prior disease severity and which outlasts the half-life of the affected innate immune cell subsets.

## Figures and Tables

**Figure 1 cells-13-00300-f001:**
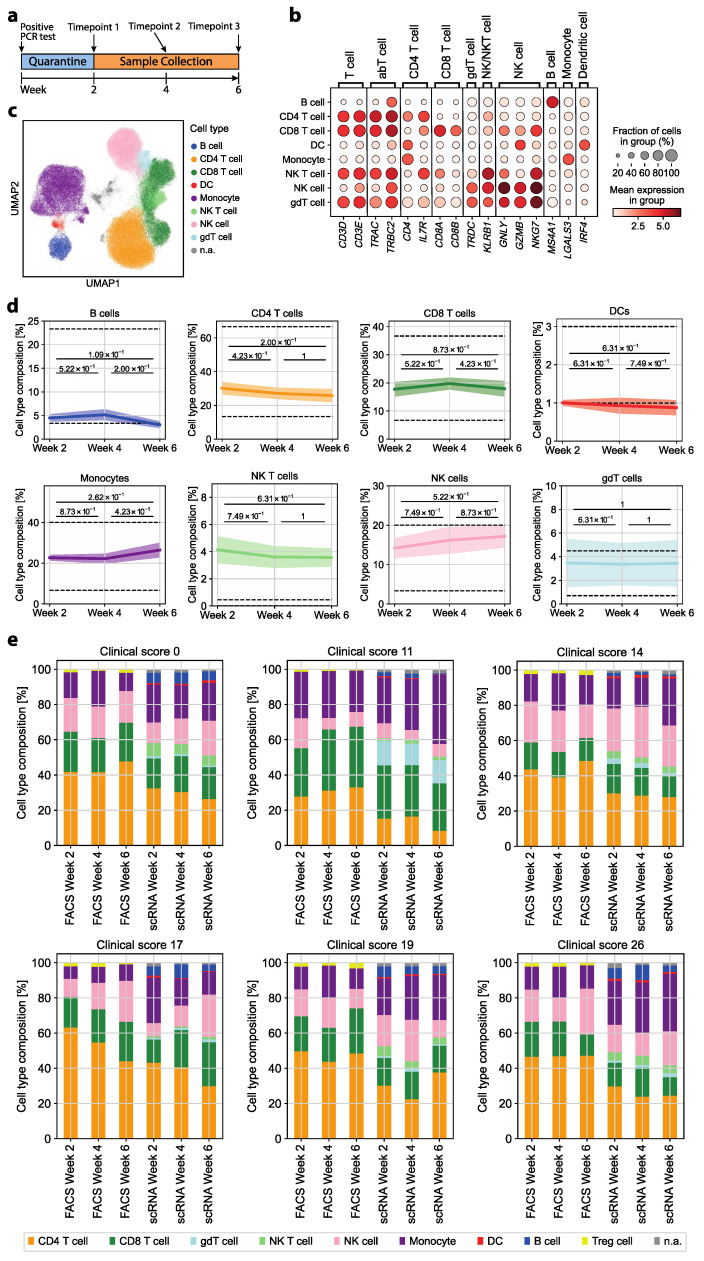
Convalescent COVID-19 patients show longitudinal stability in the relative frequency of immune cell types from the blood. (**a**) Timeline of patient sample collection. (**b**) Mean log-normalized marker gene expression for each of the assigned cell types and not annotated (n.a.) cells. (**c**) UMAP representation of all cells after preprocessing with coloring of the assigned cell type for each cell. A total of 1345 cells were n.a. due to ambiguous marker gene expression. (**d**) Relative cell type frequency among all cells over time. The mean relative frequency of all patients ± standard error is shown. For each pair of time points, a two-sided Wilcoxon rank sum test was performed. The dashed line indicates the range of physiological variation in the relative cell type frequency for the respective cell type. (**e**) Cell type composition of each patient over time based on marker gene annotation for scRNA-seq or differential gating by flow cytometry.

**Figure 2 cells-13-00300-f002:**
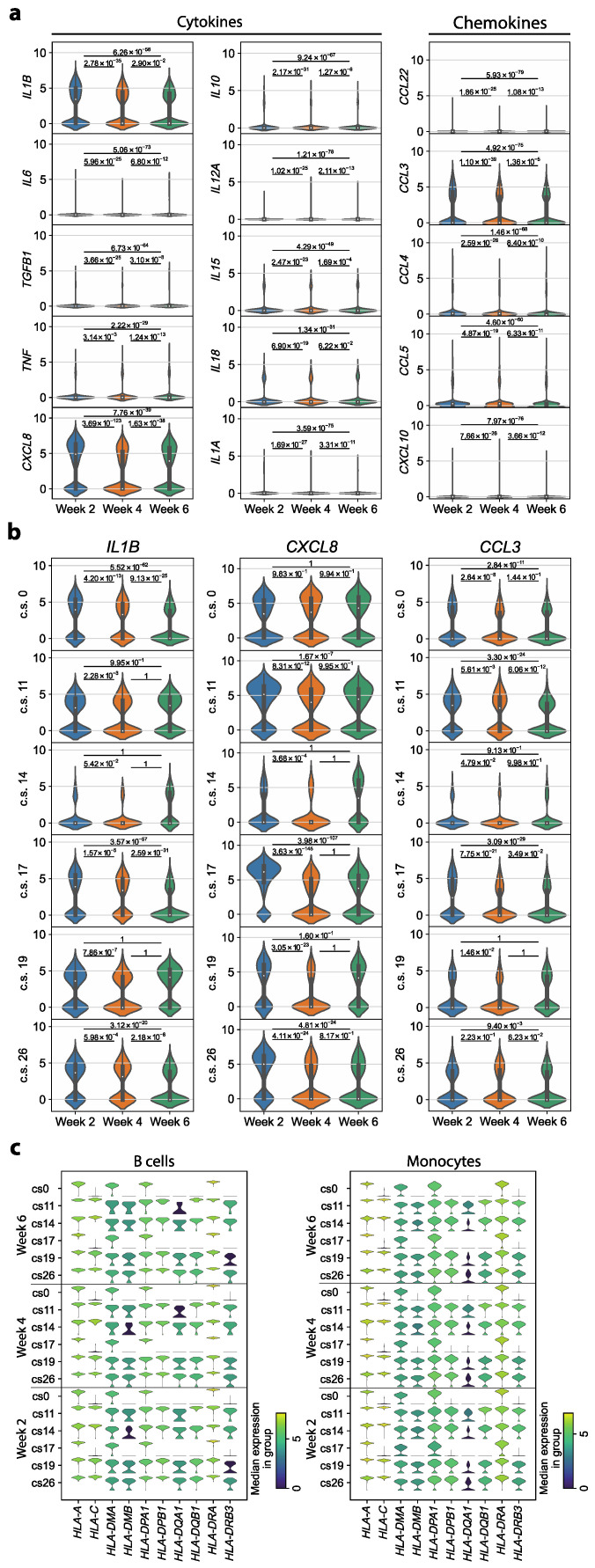
Proinflammatory immune parameters of immune perturbation in acute COVID-19 do not persist in convalescent COVID-19 patients. (**a**) Log-normalized expression of cytokines and chemokines that are known to be expressed in monocytes. Shown is the expression distribution in monocytes from all patients, ordered by time points. For each pair of time points, a two-sided Wilcoxon rank sum test was performed. (**b**) Log-normalized expression of *IL1B*, *CXCL8* and *CCL3* in the monocytes of each patient individually, ordered by time point. Each patient is represented by the respective clinical score (c.s., range 0–45). *p* values refer to a Wilcoxon rank sum test with the alternative hypothesis that the expression at the respective earlier time point is greater. (**c**) Log-normalized expression of MHC class I and II molecules in B cells and monocytes. Shown is one violin plot per time point and patient, represented by the respective c.s.

**Figure 3 cells-13-00300-f003:**
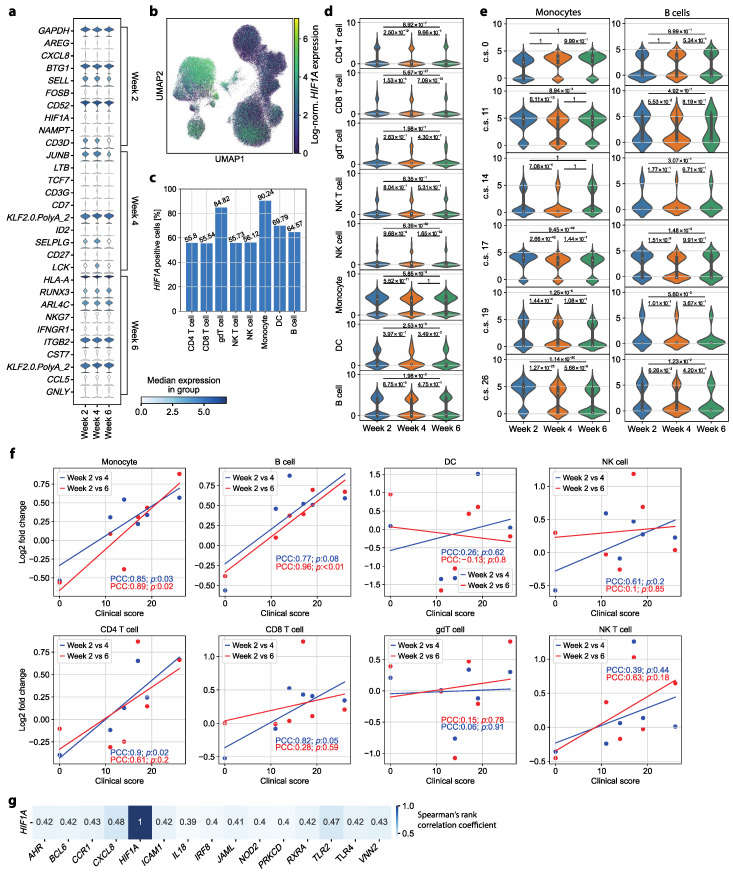
HIF1A serves as a severity-sensitive immunological scar in convalescent COVID-19 patients. (**a**) Top ten upregulated genes per time point compared to the samples for the other two time points combined. Cells from all cell types and all patients were considered. For each time point, the genes are ordered according to their z score after performing a Wilcoxon rank sum test. The top gene at each time point indicates the gene with the highest z score. (**b**) Log-normalized *HIF1A* expression per cell colored in the UMAP representation of all cells (all cell types, all patients). (**c**) Percentage of *HIF1A*^+^ cells for each cell type. A cell was considered *HIF1A*^+^ if the log-normalized *HIF1A* expression of that cell was greater than zero. (**d**) Log-normalized *HIF1A* expression in each cell type, ordered by time point. For each cell type, the cells of all patients were considered. For each pair of time points, we performed a Wilcoxon rank sum test with the alternative hypothesis that the expression at the respective earlier time point is greater than the expression at the later time point. (**e**) Log-normalized *HIF1A* expression in monocytes of each patient, represented by the respective clinical score (c.s., range 0–45), ordered by time point. (**f**) log2FC (*y*-axis) of the mean log-normalized *HIF1A* expression in each cell type in week 2 compared to week 4 (blue) and week 6 (red) depending on the clinical score of the patient (*x*-axis). The trend of the data points is indicated as a line (linear regression). The strength of the respective correlation is described by the Pearson correlation coefficient (PCC) and *p* value (*p*). (**g**) Genes co-expressed with *HIF1A*. Cells of all cell types and patients were considered. Only genes with a Spearman’s rank correlation greater than 0.39 or smaller than −0.39, resulting in the top 15 correlated genes, are shown.

**Figure 4 cells-13-00300-f004:**
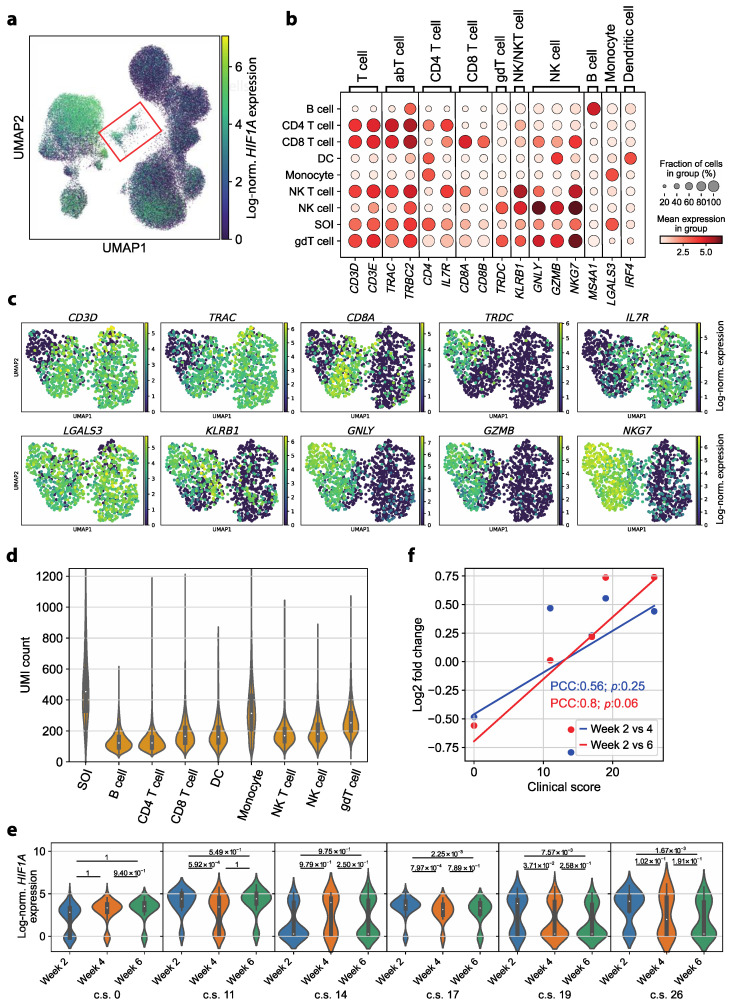
Circulating complexes formed by monocytes with either T cells or NK cells represent a persisting dysregulation in convalescent COVID-19 patients. (**a**) Log-normalized *HIF1A* expression per cell colored in the UMAP representation of all cells (all cell types, all patients, all time points). The subcluster of interest (SOI) is marked by a red box. (**b**) Mean log-normalized expression of marker genes in each cell type and the subcluster of interest (SOI). (**c**) Log-normalized expression of selected marker genes in the UMAP representation of the SOI after calculating a neighbor graph. (**d**) UMI count per cell in each cell type and the SOI. The SOI depicts significantly higher UMI counts than the monocytes (*p* value of Wilcoxon rank sum test with the alternative hypothesis that the UMI counts in the SOI are greater: 4.36 × 10^−123^). (**e**) Log-normalized *HIF1A* expression in the cells in the subcluster of interest for each patient, represented by the respective clinical score (c.s., range 0–45), ordered by time point. A Wilcoxon rank sum test with the alternative hypothesis that the expression at the respective earlier time point is greater than the expression at the later time point was performed for each pair of time points. (**f**) Log2-fold change (*y*-axis) of the mean log-normalized *HIF1A* expression in the SOI in week 2 compared to week 4 (blue) and week 6 (red) depending on the c.s. of the patient (*x*-axis). The trend of the data points is indicated by a linear regression (line). The strength of correlation is described by the Pearson’s correlation coefficient (PCC) and *p* value (*p*).

## Data Availability

All data is publicly available through the COVID-19 Cell Atlas, where they can be interactively explored: https://www.covid19cellatlas.org/ (accessed on 29 January 2024). The code for the data analysis is publicly deposited on GitHub and is accessible at the following link: https://github.com/Lilly-May/HIF1A-immuno-scar-COVID19 (accessed on 29 January 2024).

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
