# Peer review of "Single-Cell RNA Sequencing Reveals HIF1A as a Severity-Sensitive Immunological Scar in Circulating Monocytes of Convalescent Comorbidity-Free COVID-19 Patients"

_cells, 2024, doi:10.3390/cells13040300_

Round 1

Reviewer 1 Report

Comments and Suggestions for Authors

May et al.'s study examines single-cell transcriptomics in convalescent COVID-19 patients without comorbidities, revealing HIF1A expression in circulating monocytes as a lasting biomarker and identifying circulating monocyte complexes with T cells or NK cells expressing HIF1A as recovery-associated markers, suggesting potential implications for immune monitoring and treatment strategies. While this longitudinal study involving 6 convalescent COVID-19 patients is intriguing, well-written, and exhibits thorough data analysis and presentation, it is imperative to address certain limitations before publication, either through the inclusion of additional data and/or by conducting a reanalysis.

I have two main points to be addressed:

1. The authors acknowledge that the study involves only 6 convalescent patients analyzed at various time points, starting 2 weeks after positive SARS-CoV2 testing. To enhance its robustness, additional controls, such as patients recovering from respiratory virus infections other than COVID-19 without comorbidities, and, if possible, data from the same patients during both acute SARS-CoV2 and non-SARS-CoV2 infections. These additional data could provide valuable insights into whether the presented data specifically pertains to SARS-CoV2 or reflects a broader consequence of the immune response strength during acute infection.

2. Considering the tight regulation of HIF1a expression in response to oxygen exposure at the protein and/or mRNA level, it raises concerns about potential bias in HIF1A expression due to variations in sample processing, including the isolation of PBMCs and the capturing of single cells for transcriptomics at different times. It would be beneficial to address this concern in the manuscript, discussing the possibility of standardizing sample processing and, if feasible, reanalyzing the single-cell data for HIF1a target gene expression (gene set enrichment anaysis if possible with the targeted scRNA seq approach) to demonstrate functional expression in the relevant cell types. 

Author Response

Reviewer 1.

May et al.'s study examines single-cell transcriptomics in convalescent COVID-19 patients without comorbidities, revealing HIF1A expression in circulating monocytes as a lasting biomarker and identifying circulating monocyte complexes with T cells or NK cells expressing HIF1A as recovery-associated markers, suggesting potential implications for immune monitoring and treatment strategies. While this longitudinal study involving 6 convalescent COVID-19 patients is intriguing, well-written, and exhibits thorough data analysis and presentation, it is imperative to address certain limitations before publication, either through the inclusion of additional data and/or by conducting a reanalysis.

Response: We thank the reviewer for this positive feedback. Details on how we addresses the the limitations with new analyses are provided below.

I have two main points to be addressed:

  1. The authors acknowledge that the study involves only 6 convalescent patients analyzed at various time points, starting 2 weeks after positive SARS-CoV2 testing. To enhance its robustness, additional controls, such as patients recovering from respiratory virus infections other than COVID-19 without comorbidities, and, if possible, data from the same patients during both acute SARS-CoV2 and non-SARS-CoV2 infections. These additional data could provide valuable insights into whether the presented data specifically pertains to SARS-CoV2 or reflects a broader consequence of the immune response strength during acute infection.

Response: We thank the reviewer for acknowledging our study as intriguing, well-written and thoroughly analyzed and presented. In order to enhance the robustness of the study, as suggested by the reviewer, we have included a new analysis of single-cell transcriptomic dataset of healthy controls to highlight the actual differences to patients who have recovered from COVID-19. We identified a publicly available dataset that matches our study setup in terms longitudinal patient sample collection, starting during acute infection and continuing throughout convalescence. Although that study, in contrast to our study, does not perform an in-depth characterization of the symptom severities of individual patients, it nevertheless categorizes patients into groups with moderate and severe disease courses. We used this dataset and compared the expression to the healthy controls in the study at each available timepoint. Importantly, this corroborated our findings, since only severe patients displayed a significantly upregulated HIF1A expression compared to healthy controls during convalescence. This further supports our conclusion that HIF1A acts as a severity-sensitive immunological marker in COVID-19. A corresponding description of these additional analyses was added in the results section (Section 3.4), and the results are visualized in an additional figure, provided in the supplementary material.

Although it would additionally be interesting to also assess non-SARS-Cov2 infections in the same patients to discern the specific abrogations caused to the immune system by one versus the other viral infection, there is, unfortunately, no data from the same patients during both acute SARS-CoV2 and non-SARS-Cov2 infections available.

  1. Considering the tight regulation of HIF1a expression in response to oxygen exposure at the protein and/or mRNA level, it raises concerns about potential bias in HIF1A expression due to variations in sample processing, including the isolation of PBMCs and the capturing of single cells for transcriptomics at different times. It would be beneficial to address this concern in the manuscript, discussing the possibility of standardizing sample processing and, if feasible, reanalyzing the single-cell data for HIF1a target gene expression (gene set enrichment anaysis if possible with the targeted scRNA seq approach) to demonstrate functional expression in the relevant cell types. 

Response: We thank the reviewer for the suggestion to analyze HIF1a target genes in addition to HIF1a itself. As implied by the reviewer, gene set enrichment analysis is challenging with the targeted scRNAseq approach. In fact, we noted that out of the 314 genes comprising the HIF1a target gene set (https://maayanlab.cloud), only 7 were represented by the target genes. We have nevertheless investigated all individual genes that are represented in the target panel separately and provide the data for your perusal (Figure R1). Considering the challenging interpretation of data with incomplete gene sets, we decided to not include the data in the current manuscript. But we have discussed the point raised by the reviewer in the revised version of the manuscript. Additionally, we have stressed the standardized sample processing approach. All samples were immediately frozen and then, upon completion of patient sample acquisition, analyzed simultaneously to reduce technical bias.

Reviewer 2 Report

Comments and Suggestions for Authors

Lilly May et al. conducted an investigation into immunologic signatures associated with varying disease courses, revealing HIF1A as a putative severity-sensitive long-term immunological scar in circulating monocytes of convalescent COVID-19 patients. While intriguing, the study raises several concerns that warrant attention:

  1. 1. Figure 1c shows an absence of the Treg cluster in the UMAP, contrasting with its presence in Figures 1d and 1e. An explanation for this inconsistency is crucial to address potential uncertainties in the data.

  2. 2. In Figure 2, mRNA levels of cytokines at different time points are compared, overlooking the fact that cytokines primarily function as proteins. The reliance on mRNA data raises questions about the study's methodological appropriateness, suggesting that using ELISA to measure cytokine levels in serum would provide a more accurate representation.

  3. 3. The discussion on immunological scars lacks robust validation. To establish the predictive value of HIF1A, the authors could benefit from employing either an animal model or a larger cohort of COVID-19 patients, considering the limited number of cases in the current research.

  4. 4. While HIF1A's highest positive proportion occurs in the monocyte subcluster, caution is necessary in asserting it as a severity-sensitive immunological scar in circulating monocytes of convalescent patients. A more cautious and nuanced interpretation, potentially supported by additional experiments or data, is needed to substantiate such claims.

Author Response

Reviewer 2.

Lilly May et al. conducted an investigation into immunologic signatures associated with varying disease courses, revealing HIF1A as a putative severity-sensitive long-term immunological scar in circulating monocytes of convalescent COVID-19 patients. While intriguing, the study raises several concerns that warrant attention:

  1. Figure 1c shows an absence of the Treg cluster in the UMAP, contrasting with its presence in Figures 1d and 1e. An explanation for this inconsistency is crucial to address potential uncertainties in the data.

Response: We thank the reviewer for this helpful feedback. We strongly agree that Treg cells represent an important cell subpopulation that should be explored in this context. In the revised manuscript, we have annotated this subpopulation based on marker genes (Fig. New S3d). We observe that Treg cells are distributed within the CD4+ T cell cluster as expected. Our flow cytometric analysis (Fig. 1e) demonstrates their numerically small proportion within the total CD4+ T cell population. We performed a comprehensive analysis of the Treg population from our patient cohort (Fig.NewS3). We found that Treg cells, analogous to observations for the entire CD4+ T cell population, do not significantly contribute to the upregulation of HIF1A in week 2. Furthermore, we found no evidence for a severity-sensitive HIF1A expression in this cell type. These findings have been incorporated into the respective sections in the results. We have added the additional Supplementary Figure 3 displaying all these results.

  1. In Figure 2, mRNA levels of cytokines at different time points are compared, overlooking the fact that cytokines primarily function as proteins. The reliance on mRNA data raises questions about the study's methodological appropriateness, suggesting that using ELISA to measure cytokine levels in serum would provide a more accurate representation.

Response: We absolutely agree that cytokines function as proteins. We have tested IL-1b levels in the plasma of the patients from our cohort and find increased concentrations of IL-1b at early versus late timepoints. The data are shown for your perusal (Figure R2). We have nevertheless refrained from showing cytokine levels in serum because this would not have allowed us to identify the cellular source of the cytokines and their temporal expression. We have done so previously in another study (Chu et al. Front Immunol 2021). With the methodological design chosen for the submitted manuscript, we were now able to assign all shown cytokines to the respective cell types (e.g. monocytes, B cells) and their respective timepoints of expression. If wished, we can still add the new data (Figure R2), which is also in support of our general conclusions, into the manuscript.

  1. The discussion on immunological scars lacks robust validation. To establish the predictive value of HIF1A, the authors could benefit from employing either an animal model or a larger cohort of COVID-19 patients, considering the limited number of cases in the current research.

Response: We agree that a larger cohort of COVID-19 patients would be desirable to robustly validate the value of HIF1A as a retrospective indicator of COVID-19 severity. scRNA-Seq is often performed with a limited number of patients (given the tremendous costs) throughout the scientific literature to arrive at novel target genes for further exploration in a second step. To strengthen our conclusion that HIF1A serves as an immunological scar, we have used matched patients over multiple timepoints. The role of HIF1A has also been addressed by multiple different ways to strengthen its significance (e.g. correlation with disease severity scores).

However, to further validate our findings, we sought a publicly available dataset, which we analyzed for HIF1A expression (Liu C. et al. Cell 2021). Although this dataset was not suitable for performing a within-patient comparison of expression, we could show that HIF1A expression is significantly upregulated during the stage of convalescence in COVID-19 patients compared to healthy controls. Importantly, this finding was limited to patients with a severe disease course compared to those with a moderate one. Overall, this validates our findings in a larger patient cohort and additionally draws a comparison to healthy controls.

In addition to this additional analysis, we have incorporated the point raised by the reviewer to move into an animal model and to further test HIF1A in larger cohorts by addressing this in the discussion section of the manuscript.

  1. While HIF1A's highest positive proportion occurs in the monocyte subcluster, caution is necessary in asserting it as a severity-sensitive immunological scar in circulating monocytes of convalescent patients. A more cautious and nuanced interpretation, potentially supported by additional experiments or data, is needed to substantiate such claims.

Response: In line with the preceding feedback and response, we have stressed the role and significance of HIF1A as a retrospective marker for COVID-19 disease severity in multiple ways within our small patient cohort. We have further investigated the monocyte cluster to identify potential subpopulations driving the HIF1a response, but observed no cellular bias within the monocyte population. We have extensively discussed the phenomenon of continued HIF1A expression within a short lived innate immune cell population (hematopoiesis etc). We have now stressed the reviewer’s point further in the revised version of the manuscript that the interpretation of our results warrants caution (see above: limited patient cohort, need for animal model).

Round 2

Reviewer 1 Report

Comments and Suggestions for Authors

no further comments

Reviewer 2 Report

Comments and Suggestions for Authors

The authors addressed my concerns